# Effect of Freeze Crystallization on Quality Properties of Two Endemic Patagonian Berries Juices: Murta (*Ugni molinae*) and Arrayan (*Luma apiculata*)

**DOI:** 10.3390/foods10020466

**Published:** 2021-02-20

**Authors:** María Guerra-Valle, Siegried Lillo-Perez, Guillermo Petzold, Patricio Orellana-Palma

**Affiliations:** 1Laboratory of Cryoconcentration, Department of Food Engineering, Universidad del Bío-Bío, Av. Andrés Bello 720, 3780000 Chillán, Chile; maria.guerra1601@egresados.ubiobio.cl (M.G.-V.); silillo@egresados.ubiobio.cl (S.L.-P.); 2Doctorado en Ingeniería de Alimentos, Universidad del Bío-Bío, Av. Andrés Bello 720, 3780000 Chillán, Chile; 3Magíster en Ciencias e Ingeniería en Alimentos, Universidad del Bío-Bío, Av. Andrés Bello 720, 3780000 Chillán, Chile; 4Department of Biotechnology, Universidad Tecnológica Metropolitana, Las Palmeras 3360, P.O. Box, 7800003 Ñuñoa, Santiago, Chile

**Keywords:** freeze crystallization, murta, arrayan, physicochemical properties, bioactive compounds, antioxidant activity, process parameters

## Abstract

This work studied the effects of centrifugal block freeze crystallization (CBFC) on physicochemical parameters, total phenolic compound content (TPCC), antioxidant activity (AA), and process parameters applied to fresh murta and arrayan juices. In the last cycle, for fresh murta and arrayan juices, the total soluble solids (TSS) showed values close to 48 and 54 Brix, and TPCC exhibited values of approximately 20 and 66 mg gallic acid equivalents/100 grams dry matter (d.m.) for total polyphenol content, 13 and 25 mg cyanidin-3-glucoside equivalents/100 grams d.m. for total anthocyanin content, and 9 and 17 mg quercetin equivalents/100 grams d.m. for total flavonoid content, respectively. Moreover, the TPCC retention indicated values over 78% for murta juice, and 82% for arrayan juice. Similarly, the AA presented an increase over 2.1 times in relation to the correspondent initial AA value. Thus, the process parameters values were between 69% and 85% for efficiency, 70% and 88% for percentage of concentrate, and 0.72% and 0.88 (kg solutes/kg initial solutes) for solute yield. Therefore, this work provides insight about CBFC on valuable properties in fresh Patagonian berries juices, for future applications in health and industrial scale.

## 1. Introduction

In the last decades, berries have gained a lot of attention due to their attractive colors, interesting physicochemical properties, and excellent nutritional and organoleptic characteristics. Thus, berries are not only consumed for their physical appearance but also for their countless and positive effects on the consumers’ health, since these fruits have a significant source of micronutrients, phenolic compounds, and antioxidant activity that provide important beneficial effects for human health [1,2]. 

In this context, Chile has an important diversity of wild and endemic berries due to the different ecosystems of each region, since Chile presents from dry desert climate (North) to high-intensity rainfall rate and low temperatures (South) [3]. Hence, in southern-central Chile, there are various endemic berries such as maqui (*Aristotelia chilensis*), calafate (*Berberis microphylla*), Chilean strawberry (*Fragaria chiloensis ssp. chiloensis*), murta (*Ugni molinae*), and arrayan (*Luma apiculata*), and till date, murta and arrayan have been poorly studied. However, these berries have been used since ancient times as food, ingredients, colorants, and/or traditional medicines [4,5], and it opens the possibility of future scientific analysis in the fresh fruits and/or the development of studies for commercial exploitation through new processing technologies. 

Specifically, murta and arrayan are two interesting and exotic berry fruits recognized as “superfruits,” since these fruits present high levels in terms of fiber, vitamins, minerals, nutrients, and phytochemical composition [6]. Murta (also called murtilla, myrtle berry, or mutilla) is a wild Myrtaceae bush, with black/purple color and an intense sweet taste, and it grows from Talca (35°25′36″ S, 71°40′18″ W, Maule Region, central Chile) to the Palena River (41°28′18″ S, 72°56′12″ W, Los Lagos Region, southern Chile) [7]. Similarly, arrayan (also called cauchao) is other wild Myrtaceae endemic plant that grows in forests from Valparaíso (33°03′47″ S, 71°38′22″ W, Valparaiso Region, central Chile) to Aysen (45°34′12″ S, 72°03′58″ W, Aysen del General Carlos Ibáñez del Campo region, southern Chile). The fruits have nearly globular shape with characteristic aroma [8]. Moreover, these fruits present an important content of phenolic compounds (phenolic acids, anthocyanins, flavonoids, flavonols, and tannins), and thus, it contributes significantly to the high value of antioxidant activity measured for these fruits [9,10,11,12,13,14]. Therefore, murta and arrayan have excellent potential to be incorporated into the diet for daily consumption, either as fresh fruits or derived products such as yogurt, jams, jellies, purees, and fruit juices [15]. 

Additionally, different innovations have been adopted in the food industry due to the demands of consumers, every day more interested and aware, for healthy, safe, tasty, and ideal functional foods with low impact on the environment [16]. Furthermore, these innovations can be made at any stage during the food processing, and in recent years, the changes have been observed through the implementation of emerging thermal and non-thermal technologies with the intention to increase the extraction of components, efficiency, percentage of recovery, stability, and preservation of nutritional and organoleptic characteristics in each food product [17,18,19]. 

Accordingly, freeze crystallization (FC) has attracted increasing attention as emerging non-thermal technology due to their important implications for the concentration of food solutions. Specifically, FC uses low temperatures to concentrate liquid solutions and it is based on the freezing of the water, and then, the unfrozen solution (cryoconcentrated) is separated from the ice crystals (frozen fraction). Thereby, FC allows increasing the solutes and bioactive compounds in the cryoconcentrated fraction. In turn, FC had a significant energy advantage compared to the traditional concentration technologies, since FC uses only 14% of the total energy used in evaporative concentration [20]. Hence, FC has been positively applied in various liquid foods, presenting an increase in solutes and extraordinary retention of phenolic compounds and antioxidant activity [21,22,23,24,25]. However, to our best knowledge, there are a few studies available about the application of FC on valuable “superfruit” juices [26,27]. 

Thus, the objective of this study was to evaluate the use of block FC assisted by centrifugation technology on two wild endemic Chilean berry juices (murta and arrayan) in terms of physicochemical properties, process parameters, and quality characteristics.

## 2. Materials and Methods

### 2.1. Raw Materials

Fresh murta (*Ugni molinae*) and arrayán (*Luma apiculata*) were obtained from Valle Exploradores Ltda (46°37′27″ S, 72°40′36″ W, Puerto Río Tranquilo, XI Región de Aysén, Chile), with standard commercial maturity, i.e., uniform size and color, and without visual damage (injured) or being immature. Thus, the samples were transported in covered insulated boxes at Chillán (XVI Región del Ñuble, Chile) and were kept in a refrigerated chamber until their further analysis.

### 2.2. Fruit Juice Preparation

The fruits were washed with tap water to remove dust and dirt, and then subjected to manual pressure to obtain the juice, and then, the fresh juice was filtered using a nylon cloth (0.8 mm fine-mesh) to separate large fragments such as pulp, seeds, and peels from the liquid. Later, the fresh juice was stored at 4 °C until analysis or processing.

### 2.3. Freeze Crystallization (FC)

The FC process at three cycles was based on the protocol described by Orellana-Palma et al. [27] (Figure 1). First, the plastic centrifugal tubes with 45 mL of juice were isolated with foamed polystyrene. Then, the tubes with juice were frozen through axial freezing front propagation in a static freezer (model 280, M&S Consul, Sao Paulo, Brazil) overnight at −20 °C. Later, the cryoconcentrated fraction was separated from the ice fraction by centrifugation (Eppendorf 5430R, Hamburg, Germany) at 20 °C for 20 min at 4000 rpm, and thus, this procedure can be called the first cycle (C1). The cryoconcentrate juice obtained from C1 was collected. Subsequently, the C1 solution was used as the new feed solution for the second (C2) cycle, and thus, the C2 solution was used as new feed solution for the third (C3) cycle. All the cycles were performed under the same FC procedure (axial freezing front propagation at −20 °C and centrifugation conditions). Specifically, different quality properties such as physicochemical properties, total phenolic compound content, and antioxidant activity were determined in the cryoconcentrated fraction from C1, C2, and C3.

### 2.4. Physicochemical Profile

The physicochemical profile in the samples was measured based on the method described by Orellana-Palma et al. [27]. Thus, the TSS content was measured using a digital refractometer (PAL-3, Atago Inc., Tokyo, Japan) with wide range (0–93%), and the results were expressed as Brix. The pH measurements were determined with a digital pH meter (HI-2221, Hanna Instruments, Woonsocket, RI, USA). The total titratable acidity (TTA) (grams of malic acid (MA) per liter of sample, g MA/L) was obtained by titrimetric method. The density (kg/m^3^) was obtained using a pycnometer at 25 °C. The color properties were evaluated using a colorimeter (CM-5, Konica Minolta, Osaka, Japan), and the results were provided in accordance with CIELab system (L*: darkness-whiteness, a*; greenness-redness axis, and b*: blueness-yellowness axis). The total color difference (ΔE*) was calculated in accordance with Equation (1).
(1)$$\Delta E^{*}=\sqrt{{\left(L^{*}-L_{0}^{*}\right)}^{2}+\;{\left(a^{*}-a_{0}^{*}\right)}^{2}+{\left(b^{*}-b_{0}^{*}\right)}^{2}},$$
where Δ*E** is the change variation between fresh juices and cryoconcentrated samples. The subscript 0 corresponds to the initial CIELab values in the fresh juice, and L*, a*, and b* are the color properties in each cryoconcentrated sample.

### 2.5. Determination of Total Phenolic Compound Content (TPCC)

TPCC were determined by the total polyphenol content (TPC), total anthocyanin content (TAC), and total flavonoid content (TFC) assays.

TPC, TAC, and TFC assays were measured with a spectrophotometer UV/Vis (T70, Oasis Scientific Inc., Greenville, SC, USA) based on the method described by Sekizawa et al. [28], Lee et al. [29], and Zhishen et al. [30], where gallic acid (GA), cyanidin-3-glucoside (C3G), and quercetin (Q) were used for the standard curve, respectively, and the results were expressed in mg of GA equivalents (GAE) per grams (g) of dry matter (mg GAE/g d.m.), mg of C3G equivalents per grams (g) of dry matter (mg C3G/g d.m.), and mg of Q equivalents (QE) per grams (g) of dry matter (mg QE/g d.m.), respectively.

Additionally, the TPCC retention represents the TPCC retained from the fresh juice in the cryoconcentrated solution. The TPCC retention was determined by Equation (2) [31].
(2)$$\mathrm{TPCC}\;\mathrm{retention}\;\left(\%\right)=\;\;\left(\frac{C_{o}}{C_{c}}\right)\times \left(\frac{TPCC_{c}}{TPCC_{o}}\right)\times 100,$$
where C_o_ is the initial TSS; C_c_ is the TSS at each cycle in the cryoconcentrated fraction; TPCC_c_ is the value of TPC, TAC, and TFC at each cycle; and TPCC_o_ is the initial value of TPC, TAC, and TFC.

### 2.6. Determination of Antioxidant Activity (AA)

Antioxidant activity was determined by means of the 2,2-diphenyl-1-picrylhydrazyl (DPPH), 2,2’-azino-bis(3-ethylbenzothiazoline-6-sulphonic acid) (ABTS), ferric reducing antioxidant power (FRAP), and oxygen radical absorbance capacity (ORAC) assays.

DPPH, ABTS, and FRAP assays were measured with a spectrophotometer UV/Vis (T70, Oasis Scientific Inc., Greenville, SC, USA) based on the method described by Zorzi et al. [32], Garzón et al. [33], and Chen et al. [34], respectively.

ORAC assay was evaluated based on the method described by Ou et al. [35] with a multimode plate reader (Victor X3, Perkin Elmer, Hamburg, Germany). The absorbance was measured at 485 nm (λ_excitation_) and at 520 nm (λ_emission_) every 1 min for 60 min.

Trolox (T) was used for the standard curve, and the AA assays were expressed as μM Trolox equivalents (TE) per gram (g) of dry matter (μM TE/g d.m.).

### 2.7. Process Parameters in FC

#### 2.7.1. Efficiency (η)

η (%) is defined as the solutes in the cryoconcentrated fraction relative to the solutes remaining in the ice fraction. The η (%) was determined by Equation (3).
(3)$$\eta \;\left(\%\right)=\;\frac{C_{c}-C_{I}}{C_{c}}\;*\;100\%,$$
where C_c_ and C_I_ are the TSS at each cycle in the cryoconcentrated and ice fractions, respectively.

#### 2.7.2. Solute Yield (Y)

Y (kg solutes/kg initial solutes, kg/kg) represents the relation between the recovered solute mass in the initial solution and cryoconcentrated solution. The Y (kg/kg) was determined by Equation (4).
(4)$$Y\;\left(\frac{\mathrm{kg}}{\mathrm{kg}}\right)\;=\;\frac{m_{c}}{m_{0}},$$
where m_c_ is the solute mass in the cryoconcentrated fraction and m_0_ is the initial solute mass.

#### 2.7.3. Percentage of Concentrate (PC)

PC (%) represents the weight in the initial sample relative to the weight remaining in the ice fraction. The PC was determined by Equation (5).
(5)$$\mathrm{PC}\;\left(\%\right)\;=\;\frac{W_{0}-W_{i}}{W_{0}}\;\times \;100\%,$$
where W_0_ and W_i_ are the initial and final weights in the ice fraction, respectively. 

### 2.8. Statistical Analysis

The treatments were conducted in triplicate at ambient temperature (≈22 °C), and the results were presented as mean ± standard deviation. One-way analysis of variance (ANOVA) was used to the evaluation of statistical analysis and the treatment means were compared via Fisher’s least significant difference (Fisher’s LSD) test at a confidence level of 0.95 (*p* ≤ 0.05). The data were analyzed through Statgraphics Centurion XVI software (v. 16.2.04, StatPoint Technologies Inc., Warrenton, VA, USA). 

## 3. Results and Discussion

### 3.1. Physicochemical Characteristics

The physicochemical properties in the fruit juices and CBFC-treated juices are shown in Figure 2 and Table 1. Additionally, the TSS, pH, TTA, and color (L*, a*, b* and ∆E*) values presented significant differences when each BFC cycle was compared with their respective fresh juice. 

The fresh murta juice presented lower physicochemical characteristics values than those indicated by Ah-Hen et al. [36]. However, our results were higher than previous results found in the same fruit juice by Ah-Hen et al. [37]. Moreover, the fresh arrayan juice values were higher than previous values reported by Fuentes et al. [13]. The differences can be connected to the fact that the studies were carried out with fruits from different areas of the Southern Chile, and these zones are characterized by various climatic conditions, equivalent to constant rains and low temperatures throughout the year [38]. Additionally, various factors such as type/time of harvesting, ripening process, and/or the interaction genotype-environment can change the properties of the fruit and juice [39].

First, independent of fresh juice, the TSS values showed an increasing trend with increasing BFC cycles (Figure 2). Hence, in the last cycle, for murta juice (Figure 2a), the solutes reached a TSS value close to 48.2 °Brix, and for arrayan juice (Figure 2b), the TSS value was approximately 54.0 °Brix, which is comparable to an increase in 3.4 and 3.6 times, in relation to the initial TSS values (14.0 and 15.1 °Brix), respectively. In comparison with previous studies in our laboratory, the results (third cycle) presented lower concentration index value than those obtained in orange juice (5.7) [40] and apple juice (3.9) [41], but higher than those achieved in calafate juice (3.0) [27], pineapple juice (3.3) [42], and blueberry juice (2.5) [43]. The variation in TSS values is due to the different and specific characteristics (pH, TTA, and density, among others) in the fresh juice, since each characteristic interacts under various forms in the cryoconcentration process [44]. It is worth to mention that, all the studies used the same freezing and centrifugation conditions.

The pH, TTA, and CIELab (L*, a*, and b*) values are given in Table 1.

In terms of pH, the values showed a significant decrease in both fresh murta juice (pH = 3.7) and fresh arrayan juice (pH = 5.1), since the values (third cycle) had a decrease in 27% (pH = 2.4) and 25% (pH = 3.8), in comparison to the correspondent pH value of the fresh juice, respectively. By contrast, the fresh murta juice and fresh arrayan juice increased the TTA values to 2.3 and 1.6 for C1, 3.1 and 2.1 for C2, and 3.8 and 2.6 for C3, indicating an increase in 224% and 217% (in the last cycle) in relation to the initial TTA values, respectively. The opposite behavior among pH and TTA values has been observed in various cryoconcentrated food liquids such as calafate juice [27], pineapple juice [42], and sapucaia nut cake milk [45]. The pH and TTA changes have been attributed to the continuous increases in TSS at each cycle, since it provokes an increase in the organic acids of the cryoconcentrates, causing the opposite phenomenon between pH and TTA values [41].

In terms of colorimetric CIELab parameters, the color measurement is considered an indicator of food quality and can also be used indirectly in the analysis of colored components contained in fruits, particularly in berries, providing an estimation of antioxidants and polyphenolic compounds [46,47].

Thus, all the samples presented an important change during the cycles, since each parameter had significant modifications, and thus, in C3, the L* values decrease from 52 to 14 CIELab units and from 12 to 0.4 CIELab units, for fresh murta and arrayan juices, which is equivalent to a decrease in 72% and 96%, respectively, signifying that the cryoconcentrated samples were darker than the original juice. However, as cycles advanced, a progressive increase in a* and b* values was observed, with values from 4 to 35 CIELab units and from 3 to 15 CIELab units, for murta juice, and from 9 to 39 CIELab units and from 1 to 33 CIELab units, for arrayan juice, respectively, indicating that the juices had a darkish red color (most noticeable in arrayan juice), and thus, the final cryoconcentrated samples presented a very dark visual appearance with red coloration (Figure 3). These results suggested that the changes in visual color of fresh juices treated by BFC can be linked to the notable increase in the TSS concentration and total phenolic compound content at each cycle [48]. Additionally, the ΔE* between the samples (fruit juices and CBFC-treated juices) specified that the highest ΔE* values were assessed for the final cycle, since the values ranged from 13 to 50 CIELab units and from 19 to 46 CIELab units, from the first to the third cycle, for murta juice and for arrayan juice, respectively. Specifically, Krapfenbauer et al. [49] defined that a ∆E* ≥ 3.5 CIELab units denote visual differences by the consumers between food products. Therefore, our values indicate that the human eye can perceive visual differences between the original sample and each cryoconcentrated sample, since all the ∆E* values were higher than 13 CIELab units. All the data and visual color had similar trend to the results reported in various studies on FC applied to fresh juices [27,31,43].

### 3.2. Total Phenolic Compound Content (TPCC)

The levels of TPCC in the fruit juices and CBFC-treated juices are shown in Table 2.

First, fresh murta juice had TPCC values close to 6.4 mg GAE/g d.m., 4.2 mg C3G/g d.m., and 3.2 mg QE/g d.m., while fresh arrayan juice had TPCC values of approximately 20.5 mg GAE/g d.m., 8.2 mg C3G/g d.m., and 6.0 mg QE/g d.m., for TPC, TAC, and TAC, respectively. These results were lower than the ones reported by Ramirez [50], who reported the phenolic compounds content profile of six small berries from the VIII Region del Bio-Bío of Chile. However, the differences in TBCC values between the studies might be explained by the different harvest areas, since the VIII Region del Bio-Bío has dry temperate climates in summer and rainy temperate climates in winter, and thus, the temperature varies from 0 to 30 °C throughout the year, while the Southern Chile has constant rains and low temperatures throughout the year [38,51,52]. Hence, the climatic conditions and geographical characteristics contribute to various differences on fruit maturation, and in addition, other factors such as genetic and species variabilities, harvesting year, and growing season affect the phenolic compounds content in final fruits [53].

Concerning TPCC results, a similar behavior to the TSS values was observed, since a significant increase in TPC, TAC, and TFC values was detected as cycles advanced. Thus, in the last cycle, for murta juice, the values were close to 20.0 mg GAE/g d.m., 12.5 mg C3G/g d.m., and 8.6 mg QE/g d.m., and for arrayan juice, the values were approximately 65.6 mg GAE/g d.m., 24.8 mg C3G/g d.m., and 17.0 mg QE/g d.m., for the first, second, and third cycle, respectively, and thus, these values were over 2.7 times higher than the correspondent initial TPCC value. This upward trend in TPCC values post-FC has been observed in numerous food liquids such as green tea extract [21], fruit juices [22,23,24], broccoli extract [25], coffee extract [54], and wine [55]. Additionally, the retention specified a high amount of TPCC in the cryoconcentrated fraction, since the retention values (third cycle) were close to 91%, 86%, and 78%, for murta juice, and 93%, 88%, and 82%, for arrayan juice, for TPC, TAC, and TFC, respectively. The results were consistent with the values informed by Orellana-Palma et al. [41], Casas-Forero et al. [31], and Correa et al. [56], who reported TPCC retention close to 85%, 71–91%, and 90%, in apple juice, blueberry juice, and coffee extract, respectively. Therefore, the TPCC value and retention results allow corroborating that the FC technology can be visualized as a novel alternative due to the low temperatures used to concentrate, preserve, and retain an endless number of phenolic components such as polyphenols, anthocyanins, and flavonoids in the cryoconcentrated fraction [20].

### 3.3. Antioxidant Activity (AA)

Table 3 shows the AA values of the samples.

First, in both murta and arrayan juices, the AA values (μM TE/g d.m.) were approximately 33.4, 48.1, 62.6, and 21.7, and 62.0, 84.8, 92.6, and 43.4, for DPPH, ABTS, FRAP, and ORAC, respectively. These values were lower than those found by Ramirez et al. [48], Augusto et al. [57], and Rodríguez et al. [58], who studied some quality properties of native berries species from different regions of Chile, explicating that the high variability in AA values can be justified by the interaction of factors such as edaphoclimatic conditions (climate, light, temperature, and other conditions), geographical zone, cultivar/genotype, harvesting period, cultivation practice, fruit maturation, type of storage, and even, the type of juice extraction from the fruit [59].

Once the fruit juices were subjected to three CBFC cycles, an important increase in the AA values (μM TE/g d.m.) was detected, with AA values from 96.9 to 157.0 and from 167.3 to 353.1, for DPPH, from 110.6 to 250.1 and from 229.0 to 407.0, for ABTS, from 156.5 to 338.0 and from 287.0 to 590.1, for FRAP, and from 45.6 to 108.5 and from 99.8 to 221.3, for ORAC, from the C1 to C3, for murta juice, and for arrayan juice, which is equivalent to an increase over 2.1, 2.5, 2.7, and 2.8 times, in comparison to the AA values from the fresh juices, respectively. In this way, Orellana-Palma et al. [27], Samsuri et al. [60], and Casas-Forero et al. [61] also observed higher AA values post-FC than the original sample, demonstrating the positive effects of the FC on TPCC, and in turn, these components allow concentrating and preserving the antioxidant activity from the fresh juice [21].

### 3.4. Process Parameters

The 𝜂, PC, and Y results at each cycle are presented on Figure 4.

The 𝜂 (Figure 4a) showed a significant decreasing trend from C1 to C3, with values close to 76%, 72%, and 69%, and 85%, 83%, and 79%, for murta juice and for arrayan juice, respectively. The results have similar ranges established by Bastías-Montes et al. [26], Orellana-Palma et al. [41], and Zielinski et al. [62], who studied the effect of FC in various liquid samples, specifying that the 𝜂 can be related to the TSS values. Specifically, the decrease in 𝜂 values can be related to the continuous increase in TSS values in the feed solution at each cycle, since the viscosity increases as the solutes increase, and thus, the flow of solutes was slowed in the third cycle than in the previous cycles, causing a difficulty in the separation process. In addition, the high viscosity reduces the ice purity postcentrifugation (increase in TSS values in the ice fractions cycle to cycle) [43].

The PC (Figure 4a) displayed an increasing tendency as the cycles passed, with values from 70% to 81%, for murta juice, and from 78% to 88%, for arrayan juice, from the C1 to C3, respectively. The values were similar to those reported in other samples [22,40,62], suggesting that the PC values can be connected to the components in the fresh juice, since each juice has multiple components that interact differently in the FC process, and later, in the separation process, these components can facilitate or complicate the extraction of cryoconcentrates from the ice fraction [63].

The Y parameter (Figure 4b) showed a significant increasing trend as the cycles advanced, with values of approximately 0.72, 0.75, and 0.79 (kg solutes/kg initial solutes), for murta juice, and 0.82, 0.85, and 0.88 (kg solutes/kg initial solutes), for arrayan juice, for the C1, C2, and C3, respectively. The results had comparable performance that prior studies [40,41,62], associating the Y values with the recovered mass from the original mass, i.e., for each applied cycle, a high quantity of mass can be extracted and collected as a cryoconcentrated sample [64].

## 4. Conclusions

The application of CBFC allowed obtaining higher solutes than the original sample, with an intensification of the natural color of the fresh juices, leading to increase in the phenolic compounds and antioxidant capacity in the cryoconcentrates samples, since the TPCC and AA results were over 2.7 and 2.1 times higher than the correspondent initial values, respectively. Moreover, the centrifugation improves the possibility to efficiently extract more cryoconcentrate from the ice fraction due to the high values in 𝜂, PC, and Y. These data suggest that FC can effectively improve the quality properties as well as visual appearance in any fruit juices. Thereby, the FC creates a unique opportunity for food industry, since this novel emerging non-thermal technology concentrates at low temperatures, and thus, various characteristics from the fresh juice can be retained in the final concentrated product. Therefore, the next challenge could be linked to FC in the production of concentrated juice rich in phenolic components and high antioxidant capacity from endemic fruits at pilot plant scale, and then at industrial scale, and in addition, the cryoconcentrated samples could be studied with a nutritional perspective.

## Figures and Tables

**Figure 1 foods-10-00466-f001:**
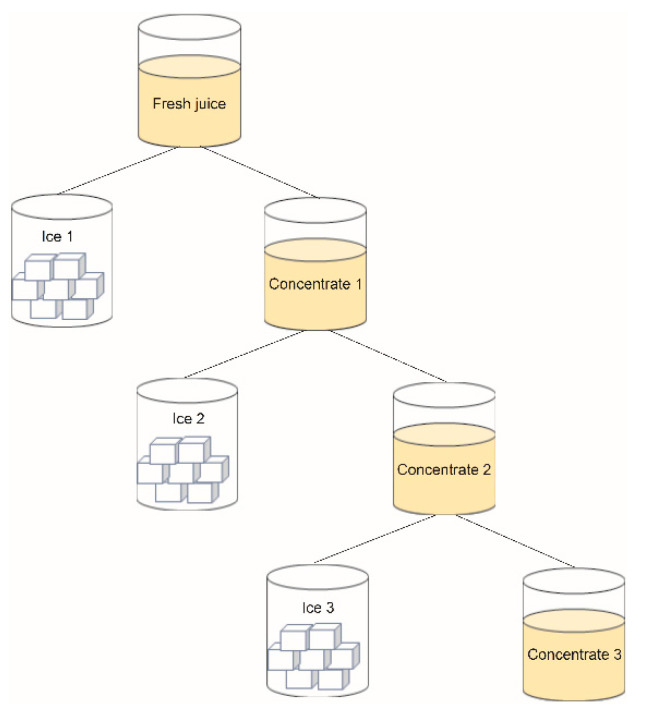
Freeze crystallization procedure at three centrifugation cycles.

**Figure 2 foods-10-00466-f002:**
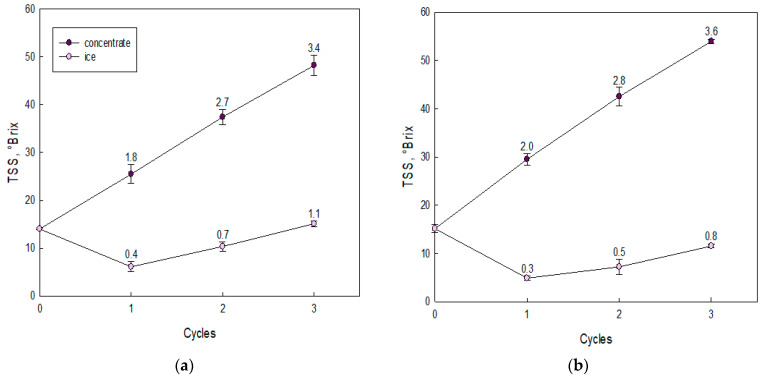
Total soluble solids (TSS) values at each BFC cycle: (**a**) murta juice and (**b**) arrayan juice. The bars represent the standard deviations. The number on the bar is the concentration index (CI), and CI is a dimensionless number that represents the increase in solutes at each BFC cycle (in both cryoconcentrated and ice fractions) with respect to the initial TSS value (C_0_) (14.0 Brix for murta juice and 15.1 °Brix for arrayan juice), i.e., CI = C_c_/C_0_, where C_c_ is the TSS value at each cycle.

**Figure 3 foods-10-00466-f003:**
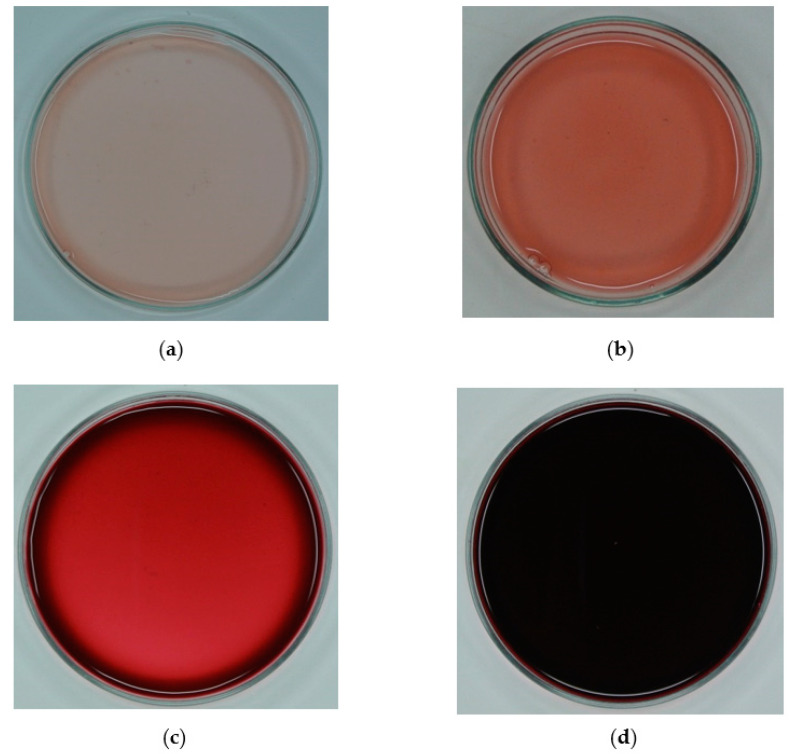
Effect on visual color of fresh murta juice and fresh arrayan juice under centrifugal block freeze crystallization (CBFC): (**a**) fresh murta juice, (**b**) cryoconcentrated murta juice, (**c**) fresh arrayan juice, and (**d**) cryoconcentrated arrayan juice.

**Figure 4 foods-10-00466-f004:**
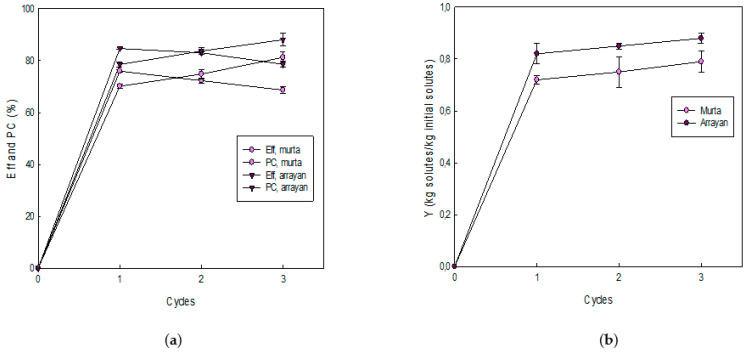
Process parameters at each CBFC cycle: (**a**) efficiency and percentage of concentrate and (**b**) solute yield.

**Table 1 foods-10-00466-t001:** Physicochemical characteristics of the samples.

Sample	pH		TTA		L*		a*		b*		∆E*	
	Murta	Arrayan	Murta	Arrayan	Murta	Arrayan	Murta	Arrayan	Murta	Arrayan	Murta	Arrayan
Fresh juice	3.7 ± 0.0 ^a^	5.1 ± 0.1 ^a^	1.7 ± 0.2 ^a^	1.2 ± 0.0 ^a^	51.9 ± 1.4 ^a^	12.2 ± 0.1 ^a^	3.8 ± 0.8 ^a^	8.7 ± 0.8 ^a^	2.6 ± 0.5^a^	1.1 ± 0.2 ^a^	-	-
Cycle 1	3.3 ± 0.2 ^b^	4.7 ± 0.2 ^b^	2.3 ± 0.1 ^b^	1.6 ± 0.2 ^b^	40.6 ± 0.7 ^b^	7.5 ± 0.5 ^b^	8.2 ± 0.4^b^	22.0 ± 0.2 ^b^	5.9 ± 0.1^b^	14.0 ± 0.2 ^b^	12.5 ± 0.7 ^a^	19.1 ± 0.5 ^a^
Cycle 2	2.8 ± 0.1 ^c^	4.2 ± 0.2 ^c^	3.1 ± 0.4 ^c^	2.1 ± 0.1 ^c^	28.2 ± 0.4 ^c^	4.1 ± 0.2 ^c^	14.7 ± 0.4 ^c^	31.0 ± 1.4 ^c^	10.1 ± 0.9^c^	22.6 ± 1.3 ^c^	27.2 ± 0.9 ^b^	32.0 ± 1.1 ^b^
Cycle 3	2.4 ± 0.1 ^d^	3.8 ± 0.0 ^d^	3.8 ± 0.2 ^d^	2.6 ± 0.3 ^d^	14.4 ± 1.4 ^d^	0.4 ± 0.5 ^d^	34.9 ± 0.3 ^d^	38.9 ± 1.2 ^d^	15.1 ± 0.9^d^	33.4 ± 0.4 ^d^	50.3 ± 1.2 ^c^	45.7 ± 0.7 ^c^

a–d: Different superscript letters in the same column indicate significant differences (*p* ≤ 0.05) according to Fisher’s LSD test. L*, a*, b*, and ∆E*, are the darkness-whiteness axis, greenness-redness axis, blueness-yellowness axis, and the total color difference, respectively.

**Table 2 foods-10-00466-t002:** Total phenolic compound content of fresh juices and cryoconcentrated samples.

Sample	TPC (mg GAE/g d.m.)		TAC (mg C3G/g d.m.)		TFC (mg QE/g d.m.)	
	Murta	Arrayan	Murta	Arrayan	Murta	Arrayan
Fresh juice	6.4 ± 0.8 ^a^	20.5 ± 2.1 ^a^	4.2 ± 0.1 ^a^	8.2 ± 0.9 ^a^	3.2 ± 0.0 ^a^	6.0 ± 0.2 ^a^
Cycle 1	8.1 ± 0.2 ^b^	28.7 ± 1.1 ^b^	5.1 ± 0.9 ^b^	10.5 ± 0.5 ^b^	3.6 ± 0.1 ^b^	6.9 ± 0.2 ^b^
TPCC retention (%)	70.0 ± 2.1 ^A^	76.9 ± 0.5 ^A^	66.6 ± 1.0 ^A^	70.8 ± 0.6 ^A^	62.1 ± 0.9 ^A^	63.2 ± 0.1 ^A^
Cycle 2	13.4 ± 1.0 ^c^	47.0 ± 2.9 ^c^	8.3 ± 1.3 ^c^	17.0 ± 2.0 ^c^	5.8 ± 0.4 ^c^	11.4 ± 1.4 ^c^
TPCC retention (%)	78.6 ± 0.9 ^B^	85.9 ± 1.2 ^B^	73.1 ± 1.4 ^B^	77.9 ± 1.1 ^B^	67.8 ± 1.3 ^B^	71.1 ± 2.9 ^B^
Cycle 3	19.9 ± 1.2 ^d^	65.6 ± 3.8 ^d^	12.5 ± 1.7 ^d^	24.8 ± 1.7 ^d^	8.6 ± 0.9 ^d^	17.0 ± 2.3 ^d^
TPCC retention (%)	91.1± 1.5 ^C^	93.1 ± 0.2 ^C^	85.6 ± 2.0 ^C^	88.0 ± 0.7 ^C^	78.1 ± 2.4 ^C^	82.5 ± 1.2 ^C^

a–d: Different small letters in the same column indicate significant differences (*p* ≤ 0.05) between the fresh juice and their cycles, according to Fisher’s LSD test. A–D: Different capital letters in the same column indicate significant differences (*p* ≤ 0.05) between the TPCC retention at each cycle, according to Fisher’s LSD test. TPCC, TPC, TAC, and TFC, are the total phenolic compound content, total polyphenol content, total anthocyanin content, and total flavonoid content, respectively.

**Table 3 foods-10-00466-t003:** Antioxidant activity (μM TE/g d.m.) of fresh juices and cryoconcentrated samples.

Sample	DPPH		ABTS		FRAP		ORAC	
	Murta	Arrayan	Murta	Arrayan	Murta	Arrayan	Murta	Arrayan
Fresh juice	33.4 ± 3.7 ^a^	62.0 ± 7.4 ^a^	48.1 ± 7.6 ^a^	84.8 ± 9.9 ^a^	62.6 ± 5.9 ^a^	92.6 ± 3.1 ^a^	21.7 ± 3.0 ^a^	43.4 ± 4.4 ^a^
Cycle 1	96.9 ± 10.1 ^b^	167.3 ± 11.6 ^b^	110.6 ± 19.1 ^b^	229.0 ± 10.4 ^b^	156.5 ± 15.7 ^b^	287.0 ± 17.2 ^b^	45.6 ± 7.4 ^b^	99.8 ± 11.7 ^b^
Cycle 2	120.2 ± 8.5 ^c^	266.4 ± 20.3 ^c^	187.6 ± 10.7 ^c^	373.1 ± 20.5 ^c^	256.7 ± 13.9 ^c^	416.6 ± 20.7 ^c^	80.3 ± 5.9 ^c^	204.0 ± 17.0 ^c^
Cycle 3	157.0 ± 12.3 ^d^	353.1 ± 14.0 ^d^	250.1 ± 17.2 ^d^	407.0 ± 7.6 ^d^	338.0 ± 21.4 ^d^	509.1 ± 23.1 ^d^	108.5 ± 10.1 ^d^	221.3 ± 20.5 ^d^

a–d: Different superscript letters in the same column indicate significant differences (*p* ≤ 0.05) according to Fisher’s LSD test.

## Data Availability

Not applicable.

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
