# Peer review of "Effect of Freeze Crystallization on Quality Properties of Two Endemic Patagonian Berries Juices: Murta (Ugni molinae) and Arrayan (Luma apiculata)"

_foods, 2021, doi:10.3390/foods10020466_

Round 1

Reviewer 1 Report

The manuscript I reviewed was mainly regarded to the food quality, but the authors did not conduct any experiment to study sensory quality. We will not want to eat food whose taste is very bad even if it is healthy. So I think including sensory data is very important to improve the manuscript.

The shown results were sufficient to say your conclusion, but the quality of taste and flavor are also one of the most important food function. You should add the sensory evaluation data to your manuscript. 

Author Response

RESPONSE TO REVIEWER COMMENTS: Reviewer 1

The manuscript I reviewed was mainly regarded to the food quality, but the authors did not conduct any experiment to study sensory quality. We will not want to eat food whose taste is very bad even if it is healthy. So I think including sensory data is very important to improve the manuscript.

The shown results were sufficient to say your conclusion, but the quality of taste and flavor are also one of the most important food function. You should add the sensory evaluation data to your manuscript.

Thank you for the observation. Unfortunately, in 2020, Chillán (Chile) was negatively impacted by the COVID-19 pandemic, and from March to October, we were in total quarantine due to the infections COVID-19. In October 2020, the number of infections per day was constant, and thus, the Universities allowed postgraduate students to enter laboratories to obtain results for their theses. However, in December, the numbers of infections per day increased significantly, and thus, in January 2021, the authorities sent Chillán to total quarantine. Hence, actually, we are still in total quarantine, therefore, we cannot go to the University to work, and thus, it is not possible to perform the sensory evaluation of the samples due to the multiple restrictions derived from the COVID-19 pandemic. Of course, it will be included the sensorial evaluation in future studies of concentrated juice from endemic Patagonian berries.

Additionally, we added the curve for covid infections in Chile, and an electronic note (in Spanish) with the start of the total quarantine in Chillán in January 2021.

http://www.ladiscusion.cl/chillan-y-chillan-viejo-retroceden-a-cuarentena-total-este-sabado/

Reviewer 2 Report

Freeze crystallization is a promising technology for concentration of heat sensitive and bioactive components. Studies focused on the applicability of freeze crystallization and freeze dehydration can provide useful information for the science and for the practice, as well. Therefore manuscript foods-1104324 has a topic that can be considered as interesting for the readers of the journal. Manuscript is generally well written with a logic structure. The theoretical background and research motivations are defined well in the Introduction section. Materials and methods are given in details. Manuscript contains valuable findings. results are discussed with relevant references.

Comments and suggestions

#1: I suggest the authors to highlight why choose the colorimetric method for the analysis (quality of concentrated juice, relationship with anthocyanins etc.)

#2: In Figure 1 the for the TSS values in concentrate and ice different range was used, and it can be unclear for the readers. I suggest the authors to use a doubel y axis, for instance to make clear the presentation of data.

#3: What is the reason for decreasing tendency of efficiency depicted in Fig 3.a. Please add a short explanation in manuscript.

Author Response

RESPONSE TO REVIEWER COMMENTS: Reviewer 2

Freeze crystallization is a promising technology for concentration of heat sensitive and bioactive components. Studies focused on the applicability of freeze crystallization and freeze dehydration can provide useful information for the science and for the practice, as well. Therefore manuscript foods-1104324 has a topic that can be considered as interesting for the readers of the journal. Manuscript is generally well written with a logic structure. The theoretical background and research motivations are defined well in the Introduction section. Materials and methods are given in details. Manuscript contains valuable findings. results are discussed with relevant references.

Thank you for the comments. Your comments are very important and motivating for us.

Comments and suggestions

  1. I suggest the authors to highlight why choose the colorimetric method for the analysis (quality of concentrated juice, relationship with anthocyanins etc.).

Thank you for this important suggestion. We have added lines on the importance of the colorimetric method (please see: 3. Results and Discussion, 3.1. Physicochemical characteristics, red letters). Moreover, two reference were added in the manuscript (please see: References, red letters). Additionally, all the references were changed in the manuscript and references section.

  1. In Figure 1 the for the TSS values in concentrate and ice different range was used, and it can be unclear for the readers. I suggest the authors to use a doubel y axis, for instance to make clear the presentation of data.

Thank you for the observation. We have made a graph with two Y axis. However, the graph looks out of the ordinary, and thus, the sense of the typical curves of solutes in freeze crystallization has been lost, in which the curves start from the initial TSS value (fresh juice, 0 in X axis), and then, the curves continue with their solutes in the cryoconcentrated fraction and ice fraction, and in addition, the graph allows adding the values of the concentration index in each cycle. In this way, we have not changed the graph, but we have added additional information in the figure legend, and thus, the graph is more easy to understand (please see: Figure 1).

  1. What is the reason for decreasing tendency of efficiency depicted in Fig 3.a. Please add a short explanation in manuscript.

Thank you for the suggestion. We have added a short explanation on the decrease in the efficiency in the manuscript (please see: 3. Results and Discussion, 3.4. Process parameters, red letters).

Reviewer 3 Report

The paper titled “Effect of freeze crystallization on quality properties of two endemic Patagonian berries: murta (Ugni molinae) and arrayan (Luma apiculata)” deals within the scope of the Foods Journal, by investigating an interesting topic of research. Please find below some remarks to help the revision of the manuscript.

Corrections to be made:

Title: The authors should point out that juices were analysed, not berries.

Lines 20-22: Please rephrase this sentence to be more understandable. Values for TPC and PC should be clearly referred to individual berries (murta and arrayan).

Line 25: Same as previous.

Line 196: Which least significant difference (LSD) test was used?

Line 242 (Figure 1): Next to the figure, authors should state what the bars and the values above them mean. Are these standard deviations or standard errors? Furthermore, please check the figure as the values do not match the size of the bars.

Lines 316-326: Reading this part and looking at the Table 2 makes me very suspicious about the presented TPCC results. It seems impossible that TPPC is increasing after each FC cycle. It is expected that they decrease as the results are expressed as grams on dry matter. It is more likely that results were obtained per 100 ml of juice. If so, authors should present the results in both ways (per g d.m. and per 100 ml) and adapt the discussion accordingly. Abstract and conclusions also need to be revised.

Line 334 (Table 2): Missing units for TPC, TAC and TFC.

Lines 339-360 and Table 3: Same as for the TPCC results. It is expected that antioxidant activity decreases as the results are expressed as grams on dry matter.

Author Response

RESPONSE TO REVIEWER COMMENTS: Reviewer 3

The paper titled “Effect of freeze crystallization on quality properties of two endemic Patagonian berries: murta (Ugni molinae) and arrayan (Luma apiculata)” deals within the scope of the Foods Journal, by investigating an interesting topic of research. Please find below some remarks to help the revision of the manuscript.

Corrections to be made:

  1. Title: The authors should point out that juices were analysed, not berries.

Thank you for the observation. The title was changed to “Effect of freeze crystallization on quality properties of two endemic Patagonian berries juices: murta (Ugni molinae) and arrayan (Luma apiculata)” (please see: tittle, red letters). Thereby, the title indicates that different berry juices were analyzed.

  1. Lines 20-22: Please rephrase this sentence to be more understandable. Values for TPC and PC should be clearly referred to individual berries (murta and arrayan).

Thank you for the observation. The abstract was restructured and rewritten, and thus, the lines are easy to understand (please see: Abstract, red letters). An important point, the abstract must contain up to 200 words, and for this reason, the abstract was restructured.

  1. Line 25: Same as previous.

Thank you for the observation. The abstract was restructured and rewritten, and thus, the lines are easy to understand (please see: Abstract, red letters). An important point, the abstract must contain up to 200 words, and for this reason, the abstract was restructured. Additionally, the improvements of the abstract were combined with comments from the reviewer 4.

  1. Line 196: Which least significant difference (LSD) test was used?

Thanks for the question, we have added more details to the line (please see: 2. Materials and Methods, 2.8. Statistical analysis, red letters). Additionally, we have specified this point in Tables 1, 2 and 3 (red letters).

  1. Line 242 (Figure 1): Next to the figure, authors should state what the bars and the values above them mean. Are these standard deviations or standard errors? Furthermore, please check the figure as the values do not match the size of the bars.

Thank you for the observation and question. The meaning of the bar and the value above them were indicated in the legend of the Figure 1 (please see: Figure 1, red letters). An important point, the number above the bars corresponds to the concentration index (CI), and CI is a dimensionless number that represents the increase in solutes at each BFC cycle (in both cryoconcentrated and ice fractions) with respect to the initial TSS value (C0) (14.0 °Brix for murta juice and 15.1 °Brix for arrayn juice), i.e., CI = Cc/C0, where Cc is the TSS value at each cycle. This dimensionless number was previously described in lines 217-218, and in turn, the definition Cs was changed to Cc (please see: 3. Results and Discussion, 3.1. Physicochemical characteristics, red letters).

An example:         Murta juice

Fresh juice    = 14.0 °Brix

Cycle 1          = 25.5 °Brix for cryoconcentrad fraction

CI (Cc/C0)      = 25.5 °Brix / 14.0 °Brix = 1.8 (please see Figure 1)

  1. Lines 316-326: Reading this part and looking at the Table 2 makes me very suspicious about the presented TPCC results. It seems impossible that TPPC is increasing after each FC cycle. It is expected that they decrease as the results are expressed as grams on dry matter. It is more likely that results were obtained per 100 ml of juice. If so, authors should present the results in both ways (per g d.m. and per 100 ml) and adapt the discussion accordingly. Abstract and conclusions also need to be revised.

Thank you for the observation. The freeze crystallization not allows the increase of the original TPCC values of fresh juice, but an important advantage of freeze crystallization is the concentration of various nutritional properties due to the concentration of solutes by the extraction of water cycle to cycle.

Now, on the results in dry matter, our research group has preferred to show the TPCC values in these units, since it is easier to determine the retention percentage. Previously, we had presented results in ml of juice (https://doi.org/10.1002/ceat.201900387 and https://doi.org/10.1590/fst.29819), but, in this moment, to make the change requested by the reviewer, we need to determine some aspects not proposed in the current study (humidity and density), and unfortunately these measurements will not be possible, since Chillán (Chile) has been in quarantine by COVID-19 from January until now (and maybe for all March). Please, see the response to reviewer 1 with more details about the quarantine by COVID-19 in Chillán. However, the comments can be used for future studies on freeze crystallization.

  1. Line 334 (Table 2): Missing units for TPC, TAC and TFC.

Thank you for the observation. We have added the units in Table 2 (please see: Table 2, red letters).

  1. Lines 339-360 and Table 3: Same as for the TPCC results. It is expected that antioxidant activity decreases as the results are expressed as grams on dry matter.

On the results in (μM TE/g d.m.), our research group has preferred to show the AA values in these units, since the results would have similar units as the TPCC values. To make the change requested by the reviewer, we need to back to the laboratories, and thus, we could determine some aspects not proposed in the current study (humidity and density), or we would have to perform all the determinations again. However, for now, it is impossible, since Chillán (Chile) has been in quarantine by COVID-19 from January until now (and maybe for all March). Please, see the response to reviewer 1 with more details about the quarantine by COVID-19 in Chillán. However, the comments can be used for future studies on freeze crystallization.

Reviewer 4 Report

Guerra-Valle M. et al. describes the effects of the freeze crystallization (FC) process at three cycles on the quality and physicochemical properties of murta juice and arrayan juice. The authors characterized wild and endemic berries found in Chile, their occurrence, composition, and health-promoting properties, and described the method of freeze crystallization. Methods of sample preparation and measurements are presented clearly and in detail. The results of many measurements: total soluble solids, pH, density, titratable acidity, color (in CIELab system), phenolic compound content, antioxidant activity, were reliably presented.

The analysis of physicochemical properties measurements and FC process parameters is consistent with the authors' main conclusion that the tested procedure can effectively improve the quality properties and the visual appearance of studied juices. The manuscript writing is fluent and easy to understand.

This article is mainly addressed at readers in the production of juices and functional foods, but I believe it may be of interest to other readers as well.

Correction:

Line 24-21: There is "Additionally…, respectively.” There is too much data in the sentence, making it difficult to understand.

Line 26 (and others): Solute yield: Is unit (kg/kg) needed?

Author Response

RESPONSE TO REVIEWER COMMENTS: Reviewer 4

  1. Guerra-Valle M. et al. describes the effects of the freeze crystallization (FC) process at three cycles on the quality and physicochemical properties of murta juice and arrayan juice. The authors characterized wild and endemic berries found in Chile, their occurrence, composition, and health-promoting properties, and described the method of freeze crystallization. Methods of sample preparation and measurements are presented clearly and in detail. The results of many measurements: total soluble solids, pH, density, titratable acidity, color (in CIELab system), phenolic compound content, antioxidant activity, were reliably presented.

Thank you for the comments. Your comments are very important and motivating for us.

  1. The analysis of physicochemical properties measurements and FC process parameters is consistent with the authors' main conclusion that the tested procedure can effectively improve the quality properties and the visual appearance of studied juices. The manuscript writing is fluent and easy to understand.

This article is mainly addressed at readers in the production of juices and functional foods, but I believe it may be of interest to other readers as well.

Thank you for the comments. Your comments are very important and motivating for us.

  1. Correction:

  • Line 24-21: There is "Additionally…, respectively.” There is too much data in the sentence, making it difficult to understand.

Thank you for the observation. The lines were rewritten, and thus, the lines are easy to understand (please see: Abstract, red letters). An important point, the abstract was rewritten, since the section must contain up to 200 words. Additionally, the improvements of the abstract were combined with comments from the reviewer 3.

  • Line 26 (and others): Solute yield: Is unit (kg/kg) needed?

Thank you for the question. Once the manuscript is accepted, the editorial office will conduct a review of the manuscript, and the office will require that all the results in the abstract present their respective units. In this way, we have made a change in the units of solute yield, since previously it was (kg/kg), and now is (kg solutes/kg initial solutes) (please see: Abstract, red letters), and thus, the unit is easy to understand. An important point, the abstract was rewritten, since the section must contain up to 200 words.

Round 2

Reviewer 3 Report

I consider the manuscript to be suitable after corrections have been made.

Author Response

Thank you for the opportunity. We are very happy because the reviewer accepted our manuscript.
